# The dynamic shape changes of the tongue base during respiration, chewing and swallowing

**Doris Haydee Rosero Salazar©, Zi-Jun Liu©\*, Amy Ly©, Yikang Dong, Alexander Veasna Simnhoung,**

Department of Orthodontics, University of Washington, Seattle, Washington, United States of America

© These authors contributed equally to this work.
\* zjliu@uw.edu

## Abstract

This study aimed to analyze dimensional deformations of the tongue base during respiration, chewing, and swallowing. Eight 7–8-month-old Yucatan minipigs were used. Under deep sedation, eight 2mm ultrasonic piezoelectric (SONO) crystals were implanted in the tongue base forming a cubic-shaped configuration, representing right/left dorsal and ventral lengths, anterior/posterior dorsal and ventral widths, and right/left anterior and posterior thicknesses. Next, 8 pairs of electromyographic (EMG) microelectrodes were inserted into the tongue, jaw, hyoid, pharyngeal, and palatal muscles. SONO and EMG signals during respiration were recorded. Then, minipigs were allowed to wake-up for unrestrained feeding. The feeding sessions were recorded with synchronized EMG and videofluoroscopy to confirm the phases of jaw movement in chewing, and swallowing episodes. Amplitudes, durations, and timings for each dimension of the SONO crystal-circumscribed region were measured from the start of the jaw opening. Findings during respiration showed elongated lengths, anterior widths and anterior thickness (p<0.05). For chewing, the width elongated up to 17% while the length and thickness shortened (12–33% and 10–32% respectively, p<0.05). Onsets of deformational changes in length and thickness occurred 10–30% earlier than in width. The cycle duration was 0.55 ± 0.11seconds chewing, and 0.69 ± 0.16seconds swallowing. During swallowing, the dorsal length (5–12%) and posterior width (10–14%) elongated whereas the posterior thickness (9–15%) and ventral length (4–10%) shortened. Explicit 3D-kinematic patterns in relation to specific functions characterize the tongue base deformation. The findings of this analysis will contribute to a better understanding of the oropharyngeal biomechanics upon abnormal conditions.

## Introduction

The tongue base, a structure located from its posterior end to the terminal sulcus and the circumvallate papilla anteriorly, seems to be anatomically, physiologically, and functionally distinct from the tongue body (located from the terminal sulcus to the tip) [1]. The tongue base regulates the openings of the larynx and esophagus during respiration, swallowing, and

**Data availability statement:** All relevant data are within the article and its supporting information files.

**Funding:** This study was supported by grant R01DE028864 from NIH/NIDCR (Z.J.L). The funder had no role in study design, data collection and analysis, decision to publish, or preparation of the manuscript.

**Competing interests:** The authors have declared that no competing interests exist.

vocalization [2,3]. Also, the movements of the tongue base seem to be synchronized with those of the jaw, soft palate, epiglottis, hyoid bone, and pharyngeal wall [4].

The implanted sonometric technique has been employed *in vivo* to investigate deformational changes and internal kinematics for real-time muscle contraction. Small SONO crystals were implanted in the muscle mass to measure the distance between crystal pairs at a determined speed of sound [5–8]. Deformational changes in thickness, length, and width were measured regardless of tissue stiffness, fat composition, or tissue fluids [5]. Previous studies on the internal kinematics of the tongue body during ingestion, drinking, and chewing indicated regional deformational changes [8–10]. For instance, variations in length, width, and thickness occur rhythmically stereotypically and synchronized with jaw movements. These studies demonstrated task-specific timings, directions (distance shortening or elongation), and amplitudes of the tongue body highly coordinated with the activities of the jaw, tongue, and hyoid muscles. However, these features in the tongue base are yet to be understood.

Our recent study in young adult minipigs using videofluoroscopy showed critical positional changes of the tongue base and other oropharyngeal structures in respiration, chewing and swallowing [11]. The major role of the tongue body in drinking, ingestion, and chewing was reported previously [10]. However, the internal kinematics of the tongue base during these functions is unknown. Therefore, the present study examined the 3D deformational changes of the tongue base in respiration and chewing/swallowing during sedated sleep and unrestrained feeding, respectively. We hypothesized that the internal kinematics of the tongue base are specific upon respiration, chewing in the pharyngeal region, and swallowing. The findings of this study are to contribute to a better understanding of the internal kinematics of the tongue base, and this will clinically support further insights into the mechanisms of respiration and swallowing disorders, such as obstructive sleep apnea (OSA) and dysphagia.

## Materials and methods

### Animals

The Institutional Animal Care and Use Committee UW reviewed and approved all experimental procedures (Protocol 3393-05), which were adhered to *the ARRIVE guidelines.* A total of healthy eight 7–8-month-old Yucatan miniature pigs (4 each sex, Premier BioSource, Ramona, CA) were used and housed one per pen. The minipigs were acclimated for 5–7 days after arrival in the new environment. Environmental enrichment and 12–24 hours dark/light cycles were provided per pen by the Animal Care Facility. Daily training on a custom-made feeding table started on day 3 and lasted until terminal experiments. The minipigs fasted overnight for up to 12–16 hours the day before recordings and their weights were monitored weekly.

### Baseline recordings

Sedation with xylazine (4mg/kg), midazolam (0.5mg/kg), and butorphanol (0.3mg/kg) was used. Additionally, maintenance with isoflurane 2–3% in oxygen was provided. This recording involved the insertions of 8 pairs of 0.10mm wire electromyographic (EMG) electrodes (California Fine Wires Co. CA) into the left tongue (genioglossus and styloglossus), palatal (tensor veli palatini and levator veli palatini), pharyngeal (middle pharyngeal constrictor), hyoid (thyrohyoid), and jaw (masseter and digastric) muscles. Only the left side was used to reduce discomfort during chewing and the stronger masseter muscle activity burst indicated ipsilateral activity [12]. A mouth mask and nasal catheter (50mm inside of left nostril) connecting with an airflow sensor (TSD160A-TSD237F, BIOPAC Co, CA) was placed for respiration and unrestrained feeding recordings respectively, as previously described [11]. The EMG and

respiratory airflow parameters were recorded using the MP150 system (BIOPAC Co, CA). The nasal catheter facilitated recording of these parameters during feeding sessions. These recordings were used to determine whether the implanted SONO crystal (Fig 1) had a significant influence on the airflow dynamics of respiration, and/or mastication in the targeted muscles.

## Surgical implantation of SONO crystals and terminal experiment

The minipig was sedated as for the baseline recording. Ringer's solution was given intravenously, and heated cushions were used for thermoregulation. Eight 2mm B-barbed SONO crystals (Sonometrics Co., Canada) were implanted through a submandibular incision to circumscribe a cubic region in the tongue base as previously described [13]. In brief, the dorsal surface remained intact, and the hyoid bone was the anatomical reference to locating the tongue base ventrally (Fig 1). The SONO crystals were positioned approximately 20mm apart, and their leading wires were sutured to the surrounding soft tissues.

After the crystal implantation, 8 pairs of EMG wire electrodes were inserted as in the baseline. The three sources of signals (SONO, EMG, and airflow) were connected and synchronized with each other via the input/output of the Biopac and Sonometric systems. Respiration was recorded for 2 minutes under sedated sleep (Fig 2B). Then, the minipig was allowed to wake up for unrestrained feeding (Fig 2C) with regular pellet mixed with barium sulfate suspension (Vet-Paque, Jorgensen Laboratories Inc. USA). To verify the jaw movements and the swallowing episodes, a synchronized x-ray videofluoroscopy (30 frames/second, GE Healthcare, OEC 9900 Elite, USA) was simultaneously taken from lateral projections (Fig 2D). Since the animal was fed unrestrainedly, the amount of food was not able to be controlled. However, the amount of ingested and chewed food should have been kept constant as a routine masticatory function. This feature was confirmed with similar bolus sizes for swallowing viewed from synchronized x-ray video fluoroscopy.

The SonoLAB software was used to record the varying distance changes of each crystal pair at a sampling rate of 150 Hz. The inspiratory/expiratory phases of respiration, jaw opening/

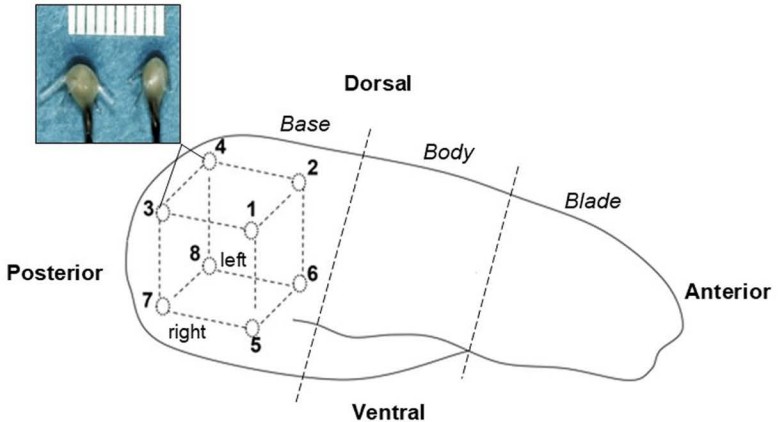

**Fig 1. Sonocrystal implantation.** The configuration of 8 ultrasonic SONO crystals (top left color image, scale 10mm) in the tongue base. Empty circles and numbers indicate the location of the crystals. Crystals #1 and #2 were implanted posterior to the two circumvallate papillae underneath the dorsal mucosa, and the #3 and #4 were implanted posterior to crystals #1 and #2 respectively and located in the dorsal area. Crystals #5, #6, #7 and #8 were placed ventrally from crystals #1, #2, #3 and #4, respectively. Each crystal was 20mm apart from each other. The selected SONO crystal pairs for length, width, and thickness are listed. The dotted lines indicate anatomical regions of the tongue. Please note the dorsal surface of the tongue base was intact as all SONO crystals were implanted via submandibular region.

closing phases of chewing, chewing side, and swallowing episodes in SONO recording were identified using the criteria previously described [11].

## Data processing and statistics

Each SONO crystal worked as a signal transmitter and receiver and the distance between each crystal pair could be measured conversely.

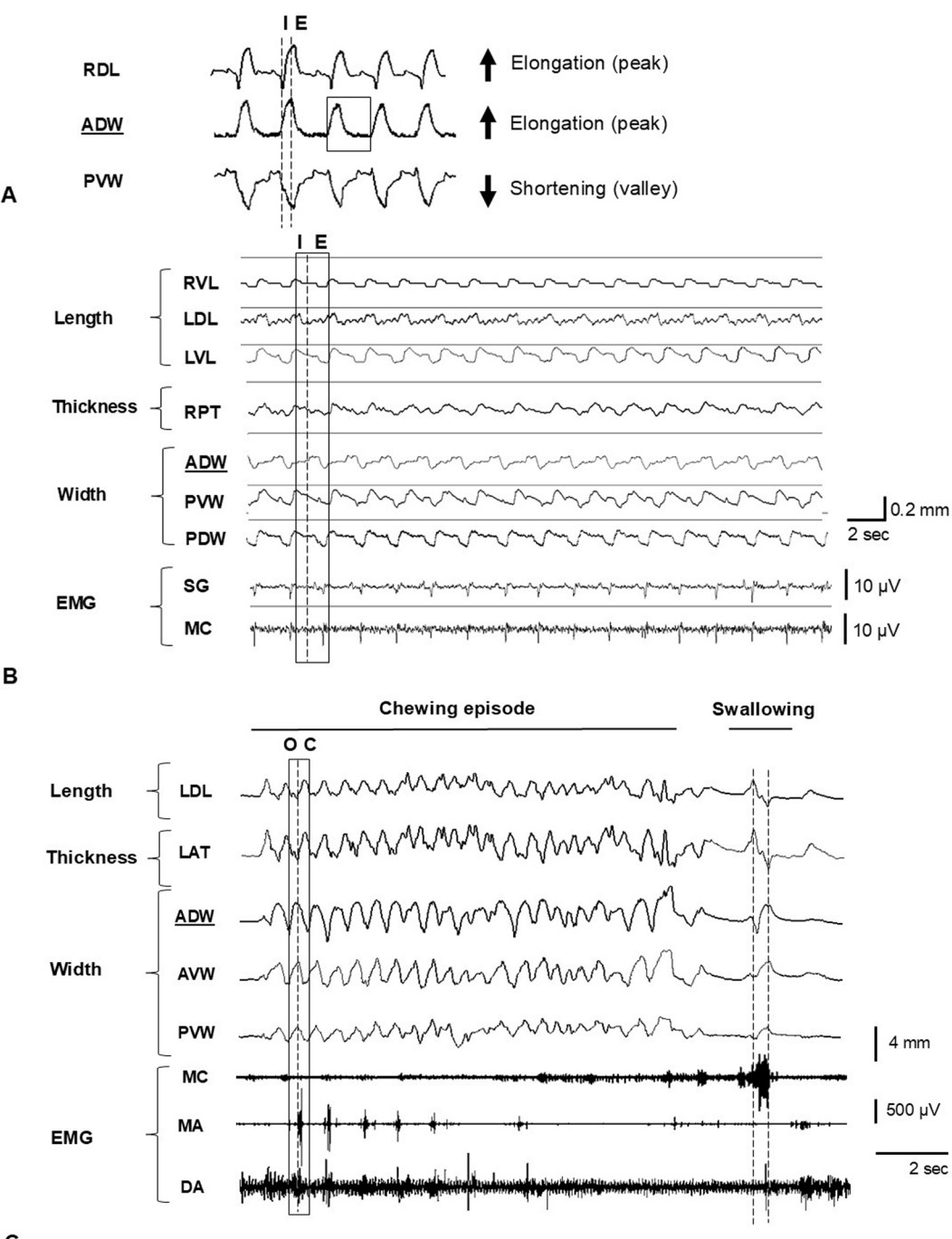

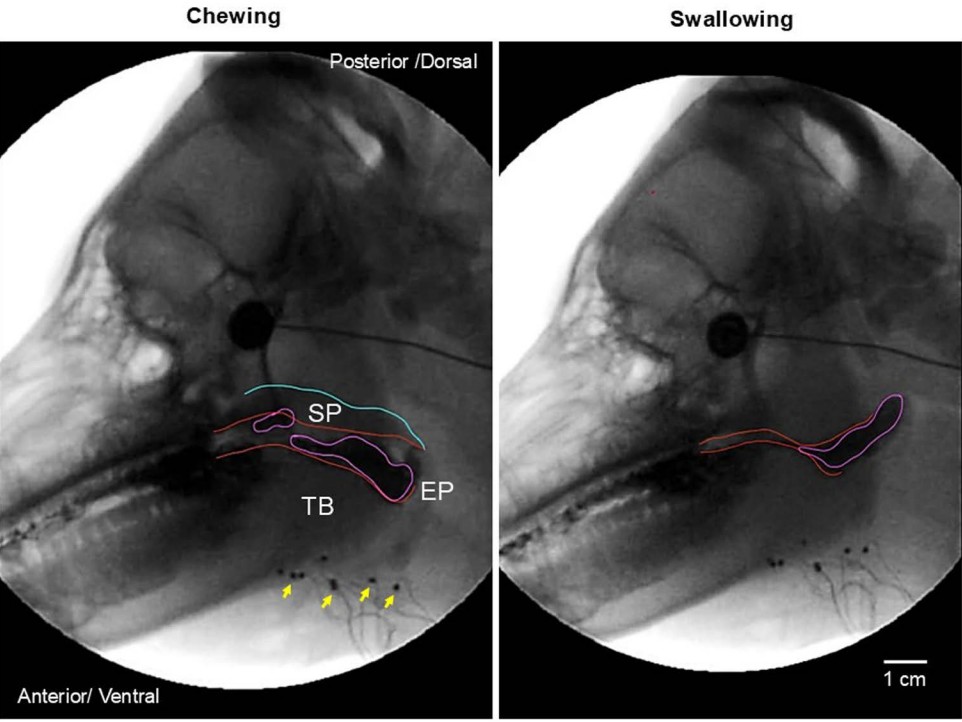

**Fig 2. Raw tracings of Sonometric recordings.** Signals obtained during respiration (A and B) and chewing/swallowing (C) alongside simultaneous x-ray fluoroscopy (D) during the feeding session. A: Sample of signals indicating the beginning of inspiration (I, dotted line), the beginning of expiration (E, dotted line), and the total respiratory cycle (box in ADW). Arrows indicate peaks or elongation (RDL, ADW) and valleys or shortening (PVW) when inspiration begins. Underlined ADW indicates the reference pair for timing analysis. B: Crystal pair signaling for length (RVL, LDL, LVL), thickness (RPT), and width (ADW, PVW, PDW). Box and dotted line indicate the respiratory cycle: inspiration (I) and expiration (E). The muscle activity (EMG) of the styloglossus (SG) and middle pharyngeal constrictor (MC) indicate small bursts throughout the cycle. C: Crystal pair signaling for length (LDL), thickness (LAT) and width (ADW, AVW, PVW). The EMG of the middle pharyngeal constrictor (MC), masseter (MA), and digastric (DA) indicate bursts in chewing during jaw opening (O, DA), jaw closing (dotted line/C and bursts in MA, DA) followed by a lower activity during swallowing with increased activity of MC (dotted lines and swallowing burst). The total chewing cycle (box) shows the beginning of jaw opening with elongation (peak) of the reference pair ADW (underline) when DA is active. The opposite occurs in jaw closing. D: Chewing (left) shows the bolus (purple lines) when processed in the back of the mouth. These events including swallowing (right) were observed using videofluoroscopy. The yellow arrows point out the location of the implanted SONO crystals. SP: soft palate. TB: Tongue base. EP: Epiglottis.

For respiration analysis, 15 stable and consecutive cycles were selected from SONO recordings for each animal. (Fig 2A). The values of the peak (upward, distance elongation) or valley (downward, distance shortening) from the baseline were computed and converted to the percentage of the initial distance values. This was to standardize the variations of the tongue sizes and distances between each crystal pair. Due to symmetrical nature of respiratory movements, all the bilateral lengths and thicknesses were combined. In addition, the time sequencing (onset) of changes in each crystal pair during inspiration was converted to a percentage of the total respiratory cycle. The crystal pair #1–2 (anterior dorsal width, ADW) presented the most available and stable wave pattern. Thus, the timing of the ADW was set at the zero-time point to be the reference for other crystal pairs in all time sequence analyses.

For chewing, 10–15 stable and consecutive chewing cycles from each animal were analyzed (Fig 2B). The computing method for distance changes was the same as for respiration. The

time sequencing analyses were the same as for respiration but were converted to a percentage of the total chewing cycle length.

Based on the real-time EMG recording using the criteria previously reported [11,12] and videofluoroscopic images, the chewing data was analyzed separately for jaw opening and closing/occlusal phases, and for the ipsilateral and contralateral chewing. An ipsilateral chewing occurred when chewing on the side corresponded to the measured crystal pairs; otherwise, was considered as contralateral chewing. Unlike chewing, swallowing episodes identified by the synchronized videofluoroscopic images occurred sporadically with no consecutive or stereotyped nature. Thus, each swallowing episode was measured independently.

SPSS (version 19.0) was used for statistical analysis. Due to the data not showing symmetrical distribution, the non-parametric Kruskal-Wallis test followed by the pair-wise multiple comparisons (u-test) were used. The significant level was set at $p < 0.05$.

## Results

Compared with the baseline data analyses, no significant differences of the terminal recordings were identified in weight, airflow and EMG, confirming that the implantation of the SONO crystals in the tongue base did not significantly affect respiration and unrestrained feeding.

For respiration, the inspiratory and expiratory phases were determined by the synchronized airflow recordings. For chewing, the jaw opening and closing/occlusal phases and chewing side were determined by the synchronized EMG and fluoroscopic recordings. For swallowing, the episodes were confirmed with synchronized barium fluoroscopic images. Due to the vulnerability of the leading wires of SONO crystals, 8 implanted crystals were not always functioning. Therefore, all measurements were collected from functional crystal pairs only. Sample sizes for animals, analyzed cycles, and available SONO crystal pairs are summarized in Table 1. The results obtained from the EMG, respiratory dynamics, and videofluoroscopic analyses were recently published elsewhere [11,14].

### Deformational dynamics in respiration

The respiratory cycles were identified as continuous movements from inspiration to expiration including post-expiratory pause (Fig 2A). The average duration of the respiratory cycle

**Table 1. Summary sample size of cycles for each pair.**

| | Crystal Pair | Total Chewing Cycles | Total swallowing episodes | Number of pigs | Total respiratory cycles | Number of pigs |
|---|---|---|---|---|---|---|
| **Length** | RDL | 11 | 2 | 1 | DL 76 | 3 |
| | LDL | 13 | 2 | 2 | | |
| | RVL | 27 | 3 | 3 | VL 30 | 1 |
| | LVL | 14 | 3 | 2 | | |
| **Thickness** | RAT | 14 | 3 | 2 | AT 45 | 2 |
| | LAT | 34 | 1 | 3 | | |
| | RPT | 13 | 1 | 2 | PT 76 | 4 |
| | LPT | 13 | 1 | 1 | | |
| **Width** | ADW | 72 | 5 | 5 | 75 | 5 |
| | PDW | 14 | 2 | 2 | 30 | 2 |
| | AVW | 45 | 3 | 3 | 30 | 2 |
| | PVW | 18 | 1 | 1 | 46 | 3 |

was 2.37 ± 0.70 seconds with inspirations of 0.80 ± 0.34 seconds and expirations/pauses of 1.56 ± 099 seconds.

As summarized in Fig 3A the timings for lengths, thicknesses, and posterior widths occurred 10–25% earlier than the reference pair (ADW). Specifically, the timing of the ventral length changes appeared 10–15% earlier than those of dorsal length, widths, and anterior thickness (p < 0.05). In contrast, the timing of the anterior ventral width (AVW) changes was largely behind that of the anterior one (ADW).

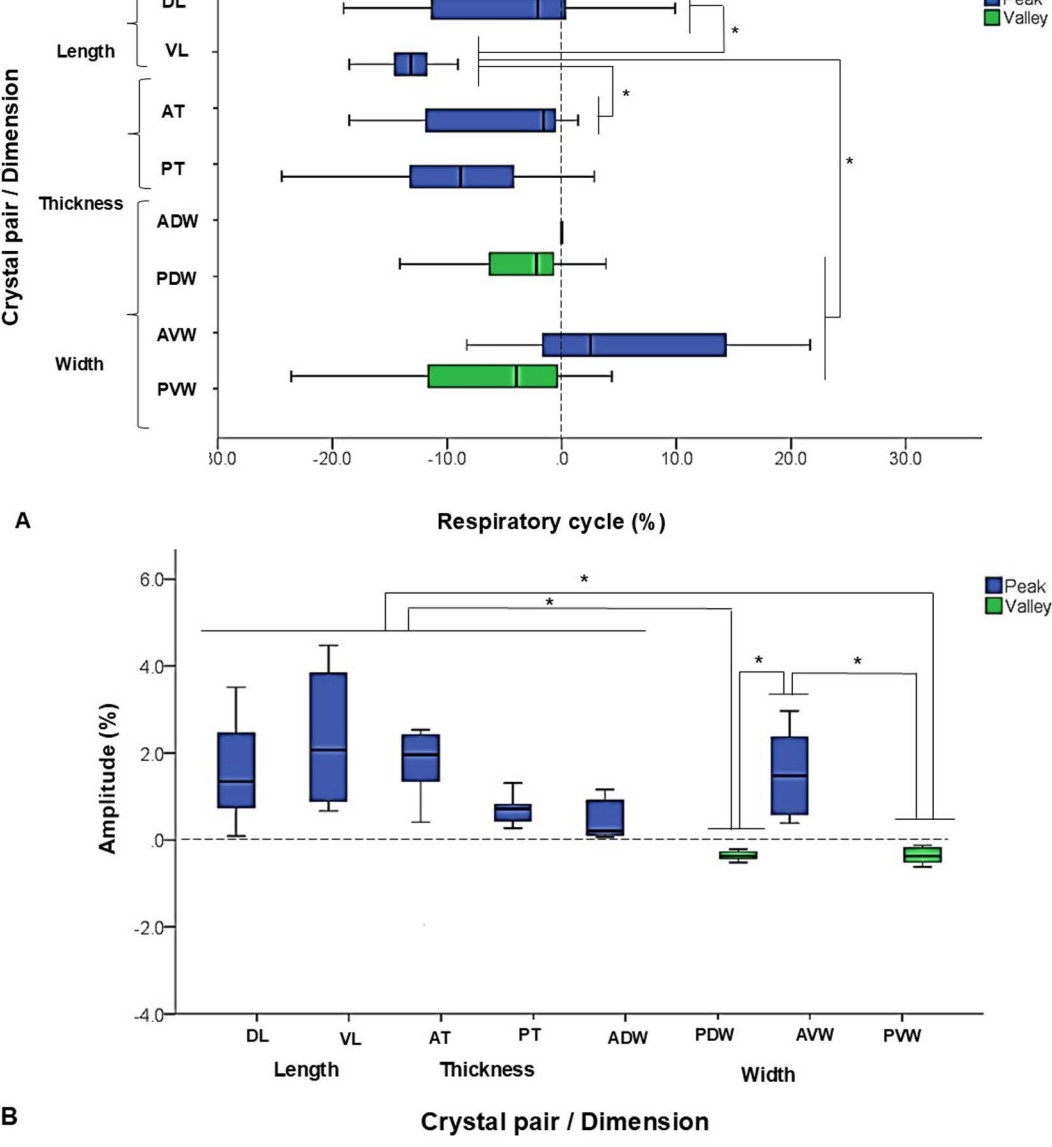

**Fig 3. Respiration.** Box-and-whisker plots show (A) time sequences and (B) amplitudes of each selected SONO crystal pair during respiratory cycles. A: % of the starting timing of the deformational change in relation to the anterior dorsal width (ADW, dotted line on time zero). *: significant difference at p < 0.05 by the non-parametric Kruskal-Wallis and pairwise test. B: % of elongation (peaks, upward waves above zero) or shortening (valleys, downward waves below zero) in relation to the initial distance in each SONO crystal pair. *: significant difference at **p** < 0.05 by the non-parametric Kruskal-Wallis and pairwise test. Refer to Fig 1 for the caption of each selected SONO crystal pair.

During inspiration, peaks were observed in lengths, thicknesses, and anterior widths. The elongation of the ventral length was larger than that dorsally whereas the increase of the anterior thickness was bigger than posteriorly. Similarly, the anterior ventral width elongated more than the dorsal one. In contrast, valleys or shortening were observed in the posterior widths.

Overall, the length, thickness, and anterior widths showed increased between 0.11–2.35%. The largest elongations was in the ventral length compared to those in thickness and anterior widths. Shortenings were between 0.30–0.71% and occurred in the posterior widths (Fig 3B). All this indicates a more elongation/widening/thickening of the anterior and ventral regions with a tendency of shortening/narrowing in the posterior and dorsal regions of the tongue base during inspiratory phases.

## Deformational dynamics in chewing

The average duration of the chewing cycle length was 0.55 ± 0.11s. During jaw opening, peaks or elongations were observed in all width dimensions. In contrast, valleys or shortenings were mostly present in lengths and thicknesses except for ventral length and posterior thickness (Fig 4). The beginning of the cycle was detected at the jaw opening phase when the synchronized burst activity from EMG recordings, i.e., activation of anterior digastric muscle, indicated elongation of the reference pair (ADW).

Overall, the ipsilateral chewing sides (0.59 ± 0.10s right and 0.57 ± 0.11s left) showed longer durations than their respective contralateral chewing side (0.57 ± 0.13s left and 0.55 ± 0.08s right).

Also, the durations of deformational changes were longer for anterior and posterior thicknesses and ventral length (LAT, RPT, and RVL: 0.60 - 0.65s), and shorter for dorsal length (RDL and LDL, 0.45 – 0.52s) (p < 0.05). No significant differences were detected between the durations of distance elongation and shortening.

The onsets for thicknesses and lengths preceded 2–40% that of the ADW (p < 0.05). Other changes in width occurred between 5–15% after that of the ADW (p < 0.05, Fig 4A). Overall, the peak waves during jaw opening usually showed onsets within the +/- 20% of that of the ADW. In contrast, the valley waves showed onsets within +/- 30% of that of the ADW (p < 0.05). This shows that dimensional elongations happen initially followed by shortenings during the jaw opening phase of chewing.

The ipsilateral and contralateral sides during chewing also showed significant differences. The onsets of the ipsilateral length (12.72 ± 6.13%) and ipsilateral thickness (7.19 ± 9.91%) occurred significantly earlier than those in their contralateral sides (2.06 ± 9.24% length, and 16.71 ± 15.61% thickness, p < 0.05). No significant differences of the onsets were found between the anterior vs. posterior, and dorsal vs. ventral regions for the changes of both lengths and thicknesses.

The range of elongation in width (ADW, PDW, AVW) was significantly larger than that of shortening in length (RDL, LDL, RVL) and thickness (RAT, LAT, p < 0.05). The dorsal lengths showed larger shortenings than those of ventral lengths. The anterior widths increased more than the posterior ones. Similarly, thickness shortened more in the anterior than posterior regions.

Additionally, the dorsal length (RDL) decreased more ipsilaterally than contralaterally (-48.46 ± 4.58% vs -30.68 ± 8.2%) whereas the thicknesses were not significantly different between ipsilateral and contralateral sides (Fig 4B). The right and left ventral lengths and posterior thicknesses showed opposite shape changes regardless of chewing side. It was observed shortening on the right length (5–20%, RVL) and thickening right posterior (5–15%, RPT) with elongation and thinning of the left length and thickness accordingly.

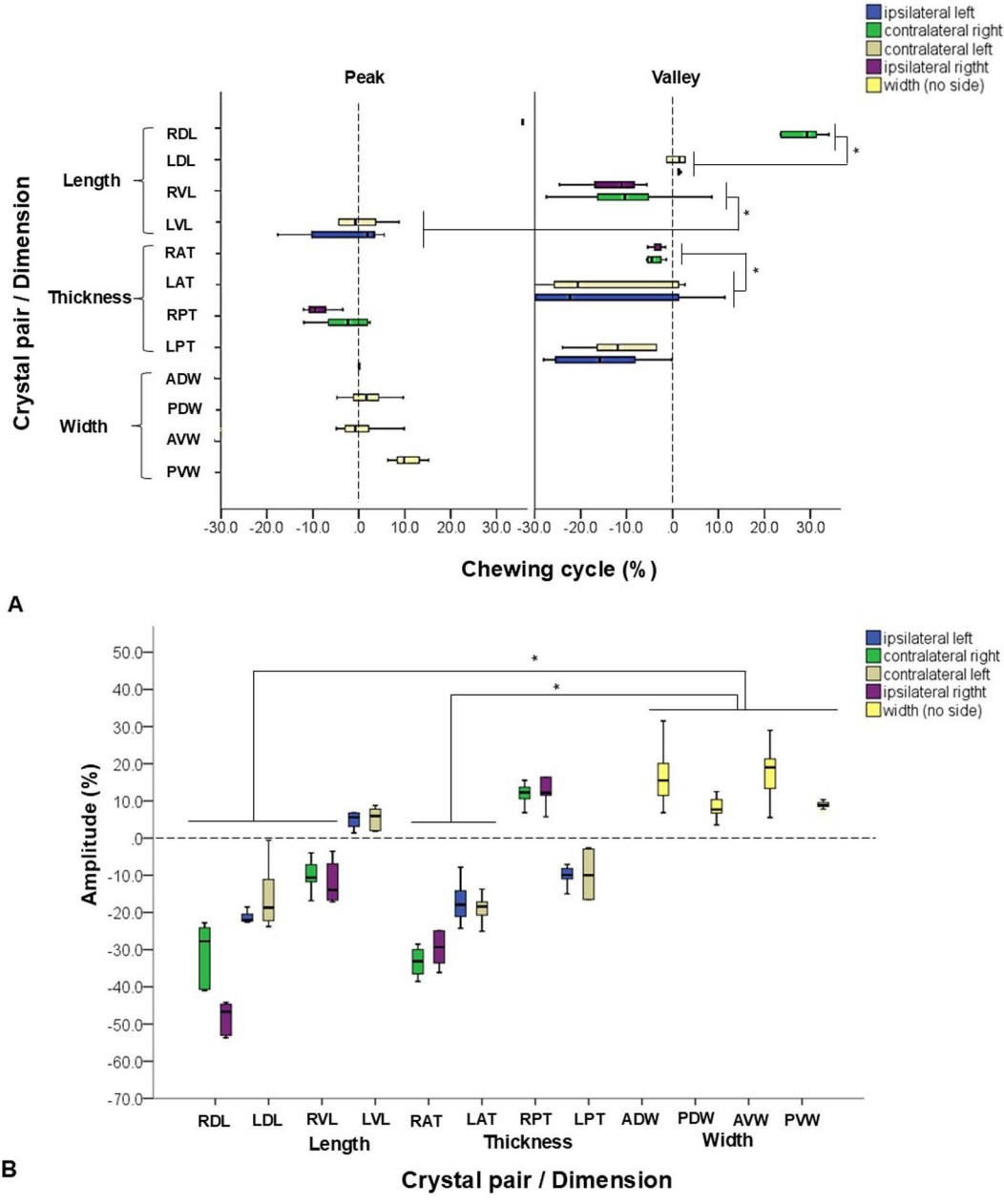

**Fig 4. Chewing.** Box-and-whisker plots show (A) time sequences and (B) amplitudes of each selected SONO crystal pair during chewing cycles. A: % of the starting timing of the deformational change in relation to the anterior dorsal width (ADW, dotted line at time zero). *: significant difference at $p < 0.05$ by the non-parametric Kruskal-Wallis and pairwise test. B: % of elongation (peaks, upward waves above zero) or shortening (valleys, downward waves below zero) in relation to the initial distance in each SONO crystal pair. *: significant difference at $p < 0.05$ by the non-parametric Kruskal-Wallis and pairwise test. Refer to Fig 1 for the caption of each selected SONO crystal pair.

## Deformational dynamics in swallowing

As shown in Fig 2B, the swallowing episode started after a brief pause of chewing cycles and occurred in jaw closing.

Given the sporadic and non-consecutive nature of swallowing, no sides were included in the analysis. A total of 10 swallowing episodes from 5 minipigs were analyzed. The average duration of the swallowing episode was 0.69 ± 0.16s. Anterior thickness, dorsal length, and dorsal width showed longer durations. In contrast, ventral width and posterior thickness had shorter durations.

The onset changes of thickness occurred 3–15% earlier than that of the anterior width (ADW). The onsets of length deformation also occurred 3–10% earlier than the ADW. On

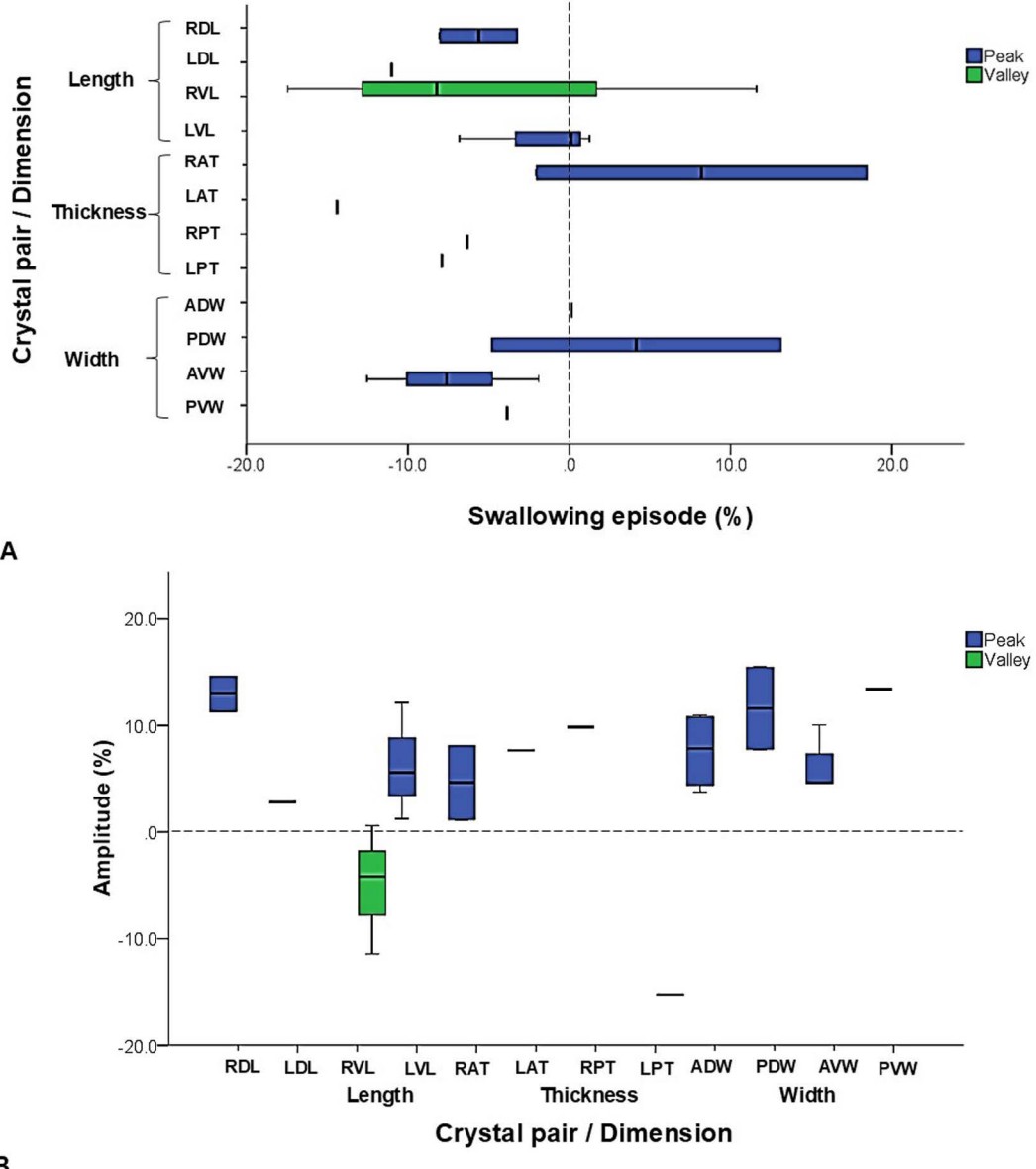

**Fig 5. Swallowing.** Box-and-whisker plots show (A) time sequences and (B) amplitudes of each selected SONO crystal pair during swallowing episodes. A: % of the starting timing of the deformational change in relation to the anterior dorsal width (ADW, dotted line at time zero). B: % of elongation (peaks, upward waves above zero) or shortening (valleys, downward waves below zero) in relation to the initial distance in each SONO crystal pair. Refer to Fig 1 for the caption of each selected crystal pair.

the other hand, the ventral widths consistently showed 3–15% earlier onsets than those of the dorsal widths. The change of the posterior dorsal width (PDW) occurred later than that of the anterior dorsal width (ADW, Fig 5A).

All distances between SONO crystal pairs elongated (peaks) with the exception of the ventral length (RVL) and posterior thickness (LPT). Although no significant differences were found, larger increases were seen in the dorsal length (RDL) and posterior widths while the largest decrease was observed in the posterior thickness (Fig 5B). This indicates dorsal elongation with posterior widening and thinning in the tongue base during swallowing. The opposite dimensional deformations in these SONO recordings indicated that swallowing movements were not constantly symmetric when it comes to the deformational changes of the tongue base dimensions.

## Discussion

### Respiration

The analysis in the present study revealed elongation of anterior lenghts, thicknesses, and widths (both dorsal and ventral) along with a minor increase in posterior thickness and simultaneous shortening of posterior widths during inspiration. Therefore, anterior lenghtening, thickening, and widening with posterior shortening of the tongue base feature the shape of the tongue base in inspiration. Some of these findings are similar to those recently reported in young adult minipigs showing lengthening and anterior dorsal widening, but enhanced deformational changes were found in obese minipigs with obstructive sleep apnea (OSA) [13]. In a recent study was analyzed the accumulation of adipose tissue in oropharyngeal structures of young adult minipigs showing tissue predominance in the tongue base and soft palate [15]. Thus, the tongue size and/or adipose tissue infiltration in obese minipigs may contribute to these changes.

Jaw muscles such as the masseter showed a weak but constant activity associated with the inspiratory phase [13]. This might be attributed to the level of muscle contraction to maintain the mandibular position (jaw tone) during inspiration. Therefore, these internal kinematics may contribute to changes of the tongue base during respiration upon volumetric alterations.

### Chewing

Dynamic deformations of the tongue base during chewing occurred in all dimensions. From jaw opening, it was predominantly seen in the dorsal length, anterior thickness, and anterior width. The first two showed the largest shortening/thinning and the last one the largest widening. The opposite should have followed during jaw closing/power stroke. As previously reported, the tongue body showed an elongated width of over 33% and a shortened length of 15–16% [10]. In the present study, the tongue base showed a 17% in widening corresponding to half of that in the tongue body. In contrast, a shortened length of 48% in the tongue base indicated a 3-fold larger deformation than that in the tongue body. All this suggests that regional deformations occur in synchrony to fulfill the functional requirements for chewing. Specifically, the larger shortening and widening of the tongue base in jaw opening and following elongation and shortening during jaw closing/occlusal phase may contribute to the formation and transportation of the bolus for swallowing.

The deformational time sequences indicate the order of the changes in each dimension of the tongue base during the chewing. The thickness and ventral length altered 10–30% earlier than the anterior dorsal width, but the changes in other widths occurred slightly later. In the tongue body study, the posterior ventral width was selected as the reference pair. The onsets for thickness and length occurred 20–50% later while the changes in width occurred earlier

[10]. All these dimensional deformations of the tongue base and body were reported by using the jaw opening phase as the beginning of the chewing cycle. Consequently, the opposite follows for the jaw closing. Thus, these results are consistent with those of a recent study in pigs reporting increased length of the tongue in jaw closing followed by shortenings in jaw opening [8]. A recent study, also analyzed deformations of the tongue body during chewing reporting transverse (left and right) and sagittal (upwards and downwards) deformations regarless of the jaw position [16]. In the present study, ventral length (RVL and LVL) and posterior thickness (RPT and LPT) showed opposite directions between left and right regarless of the chewing side (Fig 4B). This interesting feature may imply that a left tipping of the tongue base might occur during chewing, resulting in left elongation and posterior right thickening in the tongue base.

All these indicate the same dynamics in both the tongue base and body during chewing. This also suggests that directions and amplitudes of the deformational changes are specific in the tongue base and body. In addition, the time sequences of deformational changes in each dimension remained the same in the tongue base and body during chewing including thicknesses and lengths ahead of the widths. Thus, the tongue body widens first but the tongue base shortens and thins first during chewing. Therefore, it is postulated that deformational changes in the tongue base and body may not be synchronous in a real-time manner likely due to the regional neuromuscular control of the tongue.

## Swallowing

In the present study, the swallowing events followed a pause of 1.5-2.0s after multiple consecutive chewing cycles. These events appeared as a transient signal that continued with another pause of variable duration. The data indicated that the propulsive phase of swallowing is related to the dorsal lengthening, posterior widening, anterior thickening, and posterior thinning of the tongue base. These specific deformational dynamics explain the details about the internal kinematics of the tongue base when it retracts to propel the bolus passing through the oropharynx. A recent study in the tongue base of primates and humans found that the activity of the intrinsic muscles and deformational changes are likely related to the movement of the hyoid bone [17,18]. This study further found the increased width and thickness along with sequential changes in the length of the tongue base during the swallowing episodes. These changes were accompanied by the elongation of the palatoglossus, shortening of the genioglossus and the suprahyoid muscles that increased the tongue base volume for retraction. Given the similarity of the masticatory apparatus and function between pigs, primates, and humans [19,20], the present data on the internal kinematic of the tongue base during normal swallowing, provides the database for further studying of the mechanism of swallowing disorders, such as dysphagia.

## Conclusion

The deformational dynamics of the tongue base substantially vary in relation to the functional demands by playing major roles in respiration, chewing, and swallowing. The lengthening, thickening, and anterior widening contribute to the shape of the tongue base in the inspiratory phase of respiration. Ipsilateral chewing is characterized by longer durations, shorter lengths, and an increased thickness of the tongue base from the jaw opening to closing as compared with contralateral chewing. The onsets of deformational changes in various dimensions of the tongue base largely differ from those of the tongue body, specifically for the changes in the width and the length. In contrast, larger elongations occurred during swallowing and contributed to the retraction of the tongue base for the bolus propulsion over the epiglottis and towards the esophagus.

## Supporting information

**S1 File. Raw data on respiration.**
(PDF)

**S2 File. Raw data on chewing and swallowing (mastication).**
(PDF)

## Author contributions

**Conceptualization:** Zi-Jun Liu.

**Data curation:** Zi-Jun Liu, Doris Haydee Rosero Salazar, Amy Ly, Yikang Dong, Alexander Veasna Simnhoung.

**Formal analysis:** Doris Haydee Rosero Salazar, Amy Ly, Yikang Dong, Alexander Veasna Simnhoung.

**Funding acquisition:** Zi-Jun Liu.

**Investigation:** Zi-Jun Liu, Doris Haydee Rosero Salazar.

**Methodology:** Zi-Jun Liu, Doris Haydee Rosero Salazar.

**Project administration:** Zi-Jun Liu.

**Resources:** Zi-Jun Liu.

**Supervision:** Zi-Jun Liu.

**Validation:** Zi-Jun Liu, Doris Haydee Rosero Salazar, Yikang Dong, Alexander Veasna Simnhoung.

**Visualization:** Doris Haydee Rosero Salazar.

**Writing – original draft:** Doris Haydee Rosero Salazar, Amy Ly.

**Writing – review & editing:** Zi-Jun Liu, Doris Haydee Rosero Salazar.

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
