## [Decision Letter · Decision Letter 0]

14 Jan 2025

PONE-D-24-55641The Dynamics of Shape Changes in the Tongue Base during Respiration, Chewing and SwallowingPLOS ONE

Dear Dr. Liu,

Thank you for submitting your manuscript to PLOS ONE. After careful consideration, we feel that it has merit but does not fully meet PLOS ONE’s publication criteria as it currently stands. Therefore, we invite you to submit a revised version of the manuscript that addresses the points raised during the review process.

Your manuscript has been reviewed by two expert reviewers. I would like to inform you that both reviewers have not recommended publication of this manuscript in its present form. Although the manuscript presents interesting new findings, it also contains some drawbacks, as clearly described in their comments.

I would like to encourage you to revise the manuscript extensively according to their comments. You can find their comments at the end of this e-mail.

We look forward to receiving your revised manuscript.

Kind regards,

Ayako Mochizuki

Academic Editor

PLOS ONE

Journal Requirements:

NIH/NIDCR R01DE028864 to ZJL.  

The authors would like to thank Sydney Honnlee, Sophia Devore, and Elliot Willis of the

University of Washington for their help with animal experiments and data collection. This

study was supported by grant R01DE028864 from NIH/NIDCR (Z.J.L).

NIH/NIDCR R01DE028864 to ZJL.

4. In the online submission form, you indicated that the datasets generated and/or analyzed during the current study are available from the corresponding author upon reasonable request.

5. Please include your tables as part of your main manuscript and remove the individual files. Please note that supplementary tables (should remain/ be uploaded) as separate ""supporting information"" files.

Additional Editor Comments :

Your paper has been reviewed. The comments of the reviewers are included at the bottom of this letter.

The reviewers have recommended major revisions to your manuscript. Therefore, I invite you to revise and resubmit your manuscript as fast as possible.

Please carefully address the issues raised in the comments.

Kind regards,

Ayako Mochizuki

Academic Editor

PLOS ONE

Reviewers' comments:

Reviewer's Responses to Questions

**Comments to the Author**

1. Is the manuscript technically sound, and do the data support the conclusions?

Reviewer #1: Partly

Reviewer #2: Yes

2. Has the statistical analysis been performed appropriately and rigorously? 

Reviewer #1: Yes

Reviewer #2: Yes

3. Have the authors made all data underlying the findings in their manuscript fully available?

Reviewer #1: No

Reviewer #2: Yes

4. Is the manuscript presented in an intelligible fashion and written in standard English?

Reviewer #1: Yes

Reviewer #2: Yes

5. Review Comments to the Author

Reviewer #1: In this study, authors examined the deformational changes of the tongue base during respiration, mastication and swallowing in three dimensions by embedding ultrasonic piezoelectric (SONO) crystals in the tongue base of Yucatan minipigs. It is an interesting research topic, and the data could be important for understanding the mechanisms of respiration and swallowing disorders. However, it is difficult to judge the validity of the data because the quality of the figures overall is low and the explanations in the manuscript are insufficient. In order to improve these points, I think it is necessary to make significant revisions to the manuscript.

Important points

1. It is unclear how the starting timing of the peaks and valleys was determined in the waveform of the lengths, thicknesses and widths at the tongue base. In particular, the peak and valley occur repeatedly in succession during chewing. In such cases, it is important to clearly indicate where the starting timing is.

2. The resolution of the EMG waveform is very low, making it difficult to clearly identify muscle activity. In particular, in the recording of the chewing episode shown in Fig 2B, it is difficult to find muscle activity that coincides with the rhythm of chewing, i.e. the timing of the opening and closing of the jaw. Is this a typical muscle activity of the stylohyoid and masseter muscles while chewing in minipigs? According to the description in Materials and methods, it seems that this study is analyzing the deformation of the tongue base in the jaw opening phase and the jaw closing phase separately, so it is necessary to indicate some kind of indicator that shows the timing of the jaw opening phase and the jaw closing phase. Based on the following description in the manuscript, ‘The beginning of the cycle was detected at the jaw opening phase when the synchronized burst activity from EMG recordings, i.e., activation of the anterior digastric muscle, indicated elongation of the reference pair (ADW). (P12 line 7)’, I propose to show the EMG of the anterior digastric muscle at high resolution.

3. The authors recorded respiration under sedated sleep. I'm concerned that the sedation may affect the movement of the tongue base. Since chewing and swallowing can be recorded while awake, is there a reason why only respiration was recorded under sedated sleep? Please show the validity of recording under sedated sleep.

4. The manuscript contains the following sentence: ‘Sample sizes for animals, analysed cycles, and available SONO crystal pairs are summarised in Table 1.’ However, Table 1 is not included in the manuscript.

5. Fig 4 indicates that, during chewing, the RPT shows the peak wave, while the LPT shows the valley wave. Does this mean that during the jaw opening phase of chewing, regardless of whether it is on the ipsilateral or contralateral side, the thickness of the right posterior region increases, while the thickness of the left posterior region decreases? This also applies to the relationship between RVL and LVL. If the data in Figs 3-5, which are not mentioned in the manuscript, are the combined data of eight animals, then it is thought that there is a certain tendency in the deformational changes in the left and right sides of the tongue base during chewing in minipigs. Describe and discuss the morphological changes on the left and right sides during chewing.

Minor points

1. P6 line7

Is it correct to use a mouth mask to record breathing and a nasal catheter to record chewing? I wonder if the order is the other way round.

2. P11 line15 … valleys or shortening were observed in the posterior widths.

Fig. 2 seems to show that both PVW and PDW have peaks in the inspiratory phase, rather than valleys. Is this sentence in the manuscript correct?

3. P12 line13 … the durations of deformational changes were …

In relation to the starting timing, please specify how authors determined the duration.

4. Fig 1

Add a scale to the top left color image.

5. Fig 2

It is not clear which waveform the black arrow in the EMG in Figure B is pointing to. Please make this clear. In addition, indicate where the jaw opening phase is.

To clarify what is shown in the video fluoroscopy image in Figure C, indicate the names of the main structures in the image.

It is difficult to identify where the implanted SONO crystals are located within the white circle. Please indicate this more clearly.

6. Figs 3-5

The position of the box-and-whisker plots and the axis label are misaligned. IIn addition, the lines showing significant differences are also misaligned, so it is unclear which data is significant. Please arrange them.

Reviewer #2: Liu and colleagues investigated the movement of the tongue base applying ultrasonic piezoelectric (SONO) crystals. The synchronization of EMG and videofluoroscopy provided new insights into the tongue base movement during breathing, chewing and swallowing. The strength of this manuscript primarily is that it conducted movement analysis from new perspective (method) as a functional study of chewing and swallowing. It was very interesting, and I would like to comment from a procedural and clinical perspective.

Minor Comment 1: Please provide more details on the location of the SONO crystal implant. We would like more information, such as the definition of the base of the tongue.

Minor Comment 2: Please indicate whether the animal was trained or fasted before recording.

Minor Comment 3: We believe that it would be better if the definition of swallowing was based not only on SG but also on the activity of the infrahyoid muscles and swallowing apnea.

Minor Comment 4: Please write about the test food for chewing and swallowing recordings. I think that tongue movement changes depending on the amount of food eaten in one bite. What are your thoughts about this?

Minor Comment 5: Was the decision on the chewing side based just only on EMG and videofluoroscopy? Didn't you record both masseter muscles?

Minor Comment 6: Are there any differences in breathing, chewing, and swallowing behaviors between different species? Please describe any chewing or swallowing behaviors unique to mini pigs.

Minor Comment 7: Did you investigate the changes in the shape of the tongue base in response to the flow of the food bolus in the oral cavity?

Minor Comment 8: It is thought that the shape of the tongue base may change depending on the posture of the mini pig during recording, such as when it is facing down or forward. What are your thoughts on this?

Minor Comment 9: Did you investigate the changes in the shape of the tongue base during liquid drinking?

6. PLOS authors have the option to publish the peer review history of their article (what does this mean? ). If published, this will include your full peer review and any attached files.

**Do you want your identity to be public for this peer review?** For information about this choice, including consent withdrawal, please see our Privacy Policy .

Reviewer #1: No

Reviewer #2: **Yes: ** Kouta Nagoya

---

## [Author Response · Author response to Decision Letter 1]

29 Jan 2025

Responses to Journal Requirements and Reviewers’ Comments

Journal Requirements

1. When submitting your revision, we need you to address these additional requirements. Please ensure that your manuscript meets PLOS ONE's style requirements, including those for file naming. The PLOS ONE style templates can be found at

Response: The protocol for Sonocrystal implantation is based on a previous publication from our group. We added the reference accordingly on page 6. The main body and title page were revised following the formatting instructions and templates.

NIH/NIDCR R01DE028864 to ZJL.

Response: We provided accurate funding information. The sentence of Funding statement was amended to state that the funders had no role in study design, data collection and analysis, decision to publish, or preparation of the manuscript, and this statement is included in the cover letter.

The authors would like to thank Sydney Honnlee, Sophia Devore, and Elliot Willis of the

University of Washington for their help with animal experiments and data collection. This

study was supported by grant R01DE028864 from NIH/NIDCR (Z.J.L).

NIH/NIDCR R01DE028864 to ZJL.

Response: Funding information was removed from Acknowledgements and the Role of Funder Statement was added to the cover letter.

4. In the online submission form, you indicated that the datasets generated and/or analyzed during the current study are available from the corresponding author upon reasonable request.

This policy applies to all data except where public deposition would breach compliance with the protocol approved by your research ethics board. If your data cannot be made publicly available for ethical or legal reasons (e.g., public availability would compromise patient privacy), please explain your reasons for resubmission and your exemption request will be escalated for approval.

Response: Thank you for your clarification. We made available the raw data on respiration, chewing and swallowing as supplementary information.

5. Please include your tables as part of your main manuscript and remove the individual files. Please note that supplementary tables (should remain/ be uploaded) as separate ""supporting information"" files.

Response: Table 1 is now the part of the main body of the manuscript on page 10.

Additional Editor Comments :

Your paper has been reviewed. The comments of the reviewers are included at the bottom of this letter. The reviewers have recommended major revisions to your manuscript. Therefore, I invite you to revise and resubmit your manuscript as fast as possible.

Please carefully address the issues raised in the comments.

Response: We appreciate all comments from the two thorough reviewers. Please find our responses as follows:

Reviewer#1

1. Is the manuscript technically sound, and do the data support the conclusions?

Reviewer #1: Partly

Response: We appreciate the feedback. We added multiple editions throughout the manuscript to improve the technical soundness and scientific merits.

2. Has the statistical analysis been performed appropriately and rigorously?

Reviewer #1: Yes

Response: We appreciate this positive feedback.

3. Have the authors made all data underlying the findings in their manuscript fully available?

The PLOS Data policy requires authors to make all data underlying the findings described in their manuscript fully available without restriction, with rare exception (please refer to the Data Availability Statement in the manuscript PDF file). The data should be provided as part of the manuscript or its supporting information or deposited to a public repository. For example, in addition to summary statistics, the data points behind means, medians and variance measures should be available. If there are restrictions on publicly sharing data—e.g. participant privacy or use of data from a third party—those must be specified.

Reviewer #1: No

Response: We appreciate the feedback. The raw data files obtained from the respiration, chewing and swallowing recordings are available as supporting information (S1 and S2 files).

4. Is the manuscript presented in an intelligible fashion and written in standard English?

Reviewer #1: Yes

Response: We appreciate this positive feedback.

5. Comments to the Author

Please use the space provided to explain your answers to the questions above. You may also include additional comments for the author, including concerns about dual publication, research ethics, or publication ethics. (Please upload your review as an attachment if it exceeds 20,000 characters).

In this study, authors examined the deformational changes of the tongue base during respiration, mastication and swallowing in three dimensions by embedding ultrasonic piezoelectric (SONO) crystals in the tongue base of Yucatan minipigs. It is an interesting research topic, and the data could be important for understanding the mechanisms of respiration and swallowing disorders. However, it is difficult to judge the validity of the data because the quality of the figures overall is low and the explanations in the manuscript are insufficient. To improve these points, I think it is necessary to make significant revisions to the manuscript.

Response: Many thanks for these comments. We revised all figures for accuracy and quality.

Important points

1. It is unclear how the starting timing of the peaks and valleys was determined in the waveform of the lengths, thicknesses and widths at the tongue base. In particular, the peak and valley occur repeatedly in succession during chewing. In such cases, it is important to clearly indicate where the starting timing is.

Response: Thank you for the comment. Fig. 2A was added to show a sample of raw Sonometric recording which indicates the reference crystal pair (ADW, underlined) and the detection of onsets, peaks, valleys, and durations in all other crystal pairs. These standards were applied for all Sonometric signals on respiration, chewing, and swallowing.

2. The resolution of the EMG waveform is very low, making it difficult to clearly identify muscle activity. In particular, in the recording of the chewing episode shown in Fig 2B, it is difficult to find muscle activity that coincides with the rhythm of chewing, i.e. the timing of the opening and closing of the jaw. Is this typical muscle activity of the stylohyoid and masseter muscles while chewing in minipigs? According to the description in Materials and methods, it seems that this study is analyzing the deformation of the tongue base in the jaw opening phase and the jaw closing phase separately, so it is necessary to indicate some kind of indicator that shows the timing of the jaw opening phase and the jaw closing phase. Based on the following description in the manuscript, ‘The beginning of the cycle was detected at the jaw opening phase when the synchronized burst activity from EMG recordings, i.e., activation of the anterior digastric muscle, indicated elongation of the reference pair (ADW). (P12 line 7)’, I propose to show the EMG of the anterior digastric muscle at high resolution.

Response: We highly appreciate these comments. We improved the EMG signals in Figure 2C. We added the activities of the digastric muscle for the detection of jaw opening, the masseter for jaw closing and power stroke of chewing, and the middle pharyngeal constrictor for the swallowing episode. In addition, we improve the quality of the videofluoroscopy (Figure 2D) to show the sonocrystals and the bolus with more clarity.

3. The authors recorded respiration under sedated sleep. I'm concerned that the sedation may affect the movement of the tongue base. Since chewing and swallowing can be recorded while awake, is there a reason why only respiration was recorded under sedated sleep? Please show the validity of recording under sedated sleep.

Response: We are grateful for this comment. The minipig is active on awaking when not on feeding, thus it would be very difficult to measure the pure respiratory deformations of the tongue base. Therefore, the minipig was placed under sedation for the respiratory recording. Several literatures have indicated that the confounding effect of sedation and anesthesia on respiration has been proven to be minor (Oliven A, Odeh M, Geitini L, Oliven R, Steinfeld U, Schwartz AR, et al. Effect of co-activation of tongue protrusor and retractor muscles on pharyngeal lumen and airflow in sleep apnea patients. J Appl Physiol 2007; Oliven A, O'Hearn DJ, Boudewyns A, Odeh M, De Backer W, van de Heyning P, et al. Upper airway response to electrical stimulation of the genioglossus in obstructive sleep apnea. J Appl Physiol 2003;95(5):2023-9; Mak KH, Wang YT, Cheong TH, Poh SC. The effect of oral midazolam and diazepam on respiration in normal subjects. Eur Respir J 1993;6(1):42-7). On the other hand, we did record the respiratory features during chewing and swallowing, which was published elsewhere (Rosero-Salazar D., Honnlee S., and Liu Z.J.*: Tongue, palatal, hyoid and pharyngeal muscle activity during chewing, swallowing, and respiration. Arch. Oral Biol. 157: e105845, 2024).

4. The manuscript contains the following sentence: ‘Sample sizes for animals, analyzed cycles, and available SONO crystal pairs are summarized in Table 1.’ However, Table 1 is not included in the manuscript.

Response: We appreciate this comment. Table 1 is now included in the manuscript on page 10.

5. Fig 4 indicates that, during chewing, the RPT shows the peak wave, while the LPT shows the valley wave. Does this mean that during the jaw opening phase of chewing, regardless of whether it is on the ipsilateral or contralateral side, the thickness of the right posterior region increases, while the thickness of the left posterior region decreases? This also applies to the relationship between RVL and LVL. If the data in Figs 3-5, which are not mentioned in the manuscript, are the combined data of eight animals, then it is thought that there is a certain tendency in the deformational changes in the left and right sides of the tongue base during chewing in minipigs. Describe and discuss the morphological changes on the left and right sides during chewing.

Response: We appreciate such careful reading of our data. Fig. 4B do show that the ventral left and right lengths (RVL and LVL) and right and left posterior thicknesses had opposite deformational changes regardless of chewing side. These may indicate a left tipping of the tongue base during chewing, thus result in left elongation and posterior right thickening. These explanation was added to the results (p14) and discussion (p17).

Minor points

1. P6 line7

Is it correct to use a mouth mask to record breathing and a nasal catheter to record chewing? I wonder if the order is the other way round.

Response: We appreciate this comment. The mouth mask ensures light sedation using isoflurane 2-3% mixed with oxygen when the data was recorded during respiration. However, chewing and swallowing data was recorded when animal was awake and underwent unrestraint feeding. Clearly, mouth mask could not be used during feeding. Therefore, mouth mask was replaced by a nasal catheter to monitor respiratory airflow during chewing and swallowing. A short explanation about this was added on page 5.

2. P11 line15 … valleys or shortening were observed in the posterior widths.

Fig. 2 seems to show that both PVW and PDW have peaks in the inspiratory phase, rather than valleys. Is this sentence in the manuscript correct?

Response: We appreciate this comment. We improved all figures in terms of accuracy, clarity, and quality. The sentence in the manuscript is correct.

3. P12 line13 … the durations of deformational changes were …

In relation to the starting timing, please specify how authors determined the duration.

Response: Thank you very much for this comment. Figs. 2B and 2C show in boxes the total cycle lengths for respiration (2B) and chewing (2C), The middle-dotted lines indicate the beginning of expiratory (2B) and jaw closing (2C) phases. The time scale is in second.

4. Fig 1

Add a scale to the top left color image.

Response: Thank you very much for this comment. The image was enlarged to show the scale with a total of 10 mm. The note was added to the legend of this figure.

5. Fig 2

It is not clear which waveform the black arrow in the EMG in Figure B is pointing to. Please make this clear. In addition, indicate where the jaw opening phase is.

To clarify what is shown in the video fluoroscopy image in Figure C, indicate the names of the main structures in the image.

It is difficult to identify where the implanted SONO crystals are located within the white circle. Please indicate this more clearly.

Response: Thank you for the feedback. Fig. 2 was improved for all aspects mentioned. Each part of this and other figures were built separately for the better quality.

6. Figs 3-5

The position of the box-and-whisker plots and the axis label are misaligned. In addition, the lines showing significant differences are also misaligned, so it is unclear which data is significant. Please arrange them.

Response: Thank you for the feedback. The arrangements and editions were added accordingly.

Reviewer# 2

1. Is the manuscript technically sound, and do the data support the conclusions?

Reviewer #2: Yes

Response: We appreciate the positive feedback.

2. Has the statistical analysis been performed appropriately and rigorously?

Reviewer #2: Yes

Response: We appreciate the positive feedback.

3. Have the authors made all data underlying the findings in their manuscript fully available?

The PLOS Data policy requires authors to make all data underlying the findings described in their manuscr

---

## [Decision Letter · Decision Letter 1]

16 Feb 2025

PONE-D-24-55641R1The Dynamics Shape Changes of the Tongue Base during Respiration, Chewing and SwallowingPLOS ONE

Dear Dr. Liu,

Thank you for submitting your manuscript to PLOS ONE. After careful consideration, we feel that it has merit but does not fully meet PLOS ONE’s publication criteria as it currently stands. Therefore, we invite you to submit a revised version of the manuscript that addresses the points raised during the review process.

We look forward to receiving your revised manuscript.

Kind regards,

Ayako Mochizuki

Academic Editor

PLOS ONE

Journal Requirements:

Reviewers' comments:

Reviewer's Responses to Questions

**Comments to the Author**

1. If the authors have adequately addressed your comments raised in a previous round of review and you feel that this manuscript is now acceptable for publication, you may indicate that here to bypass the “Comments to the Author” section, enter your conflict of interest statement in the “Confidential to Editor” section, and submit your "Accept" recommendation.

Reviewer #1: (No Response)

Reviewer #2: All comments have been addressed

2. Is the manuscript technically sound, and do the data support the conclusions?

Reviewer #1: Yes

Reviewer #2: Yes

3. Has the statistical analysis been performed appropriately and rigorously? 

Reviewer #1: Yes

Reviewer #2: Yes

4. Have the authors made all data underlying the findings in their manuscript fully available?

Reviewer #1: Yes

Reviewer #2: Yes

5. Is the manuscript presented in an intelligible fashion and written in standard English?

Reviewer #1: Yes

Reviewer #2: Yes

6. Review Comments to the Author

Reviewer #1: The authors responded appropriately to all of my comments. The manuscript has been greatly improved and is now in good condition. I would like to suggest a very minor correction.

1. Fig 2C EMG Scale: Please insert a space between the 500 µV value and the unit.

2. P7 line19 “Respiration was recorded for 2 minutes under sedated sleep (Fig 2A). Then,the minipig was allowed to wake up for unrestrained feeding (Fig 2B) with regular pelletmixed with barium sulfate suspension (Vet-Paque, Jorgensen Laboratories Inc. USA).”

I think (Fig 2A) is a mistake for (Fig 2B), and (Fig 2B) is a mistake for (Fig 2C).

Reviewer #2: (No Response)

7. PLOS authors have the option to publish the peer review history of their article (what does this mean? ). If published, this will include your full peer review and any attached files.

**Do you want your identity to be public for this peer review?** For information about this choice, including consent withdrawal, please see our Privacy Policy .

Reviewer #1: No

Reviewer #2: No

---

## [Author Response · Author response to Decision Letter 2]

20 Feb 2025

Responses to the comments from Editors and Reviewers

Journal Requirements:

The reference list was reviewed, and it is complete and correct.

1. If the authors have adequately addressed your comments raised in a previous round of review and you feel that this manuscript is now acceptable for publication, you may indicate that here to bypass the “Comments to the Author” section, enter your conflict of interest statement in the “Confidential to Editor” section, and submit your "Accept" recommendation.

Reviewer #1: (No Response)

Reviewer #2: All comments have been addressed

Thank you for your positive evaluation.

2. Is the manuscript technically sound, and do the data support the conclusions?

Reviewer #1: Yes

Reviewer #2: Yes

Thank you for your positive evaluation.

3. Has the statistical analysis been performed appropriately and rigorously?

Reviewer #1: Yes

Reviewer #2: Yes

Thank you for your positive evaluation.

4. Have the authors made all data underlying the findings in their manuscript fully available?

Reviewer #1: Yes

Reviewer #2: Yes

Thank you for your positive evaluation.

5. Is the manuscript presented in an intelligible fashion and written in standard English?

Reviewer #1: Yes

Reviewer #2: Yes

Thank you for your positive evaluation.

6. Review Comments to the Author

Reviewer #1: The authors responded appropriately to all of my comments. The manuscript has been greatly improved and is now in good condition. I would like to suggest a very minor correction.

1. Fig 2C EMG Scale: Please insert a space between the 500 µV value and the unit.

A space was made between 500 and µV.

2. P7 line19 “Respiration was recorded for 2 minutes under sedated sleep (Fig 2A). Then,the minipig was allowed to wake up for unrestrained feeding (Fig 2B) with regular pelletmixed with barium sulfate suspension (Vet-Paque, Jorgensen Laboratories Inc. USA).”

I think (Fig 2A) is a mistake for (Fig 2B), and (Fig 2B) is a mistake for (Fig 2C).

Really appreciate the reviewer #1 to indicate these errors. These Figure citations were corrected in page 7.

---

## [Decision Letter · Decision Letter 2]

5 Mar 2025

The Dynamics Shape Changes of the Tongue Base during Respiration, Chewing and Swallowing

PONE-D-24-55641R2

Dear Dr. Zi-Jun Liu,

We’re pleased to inform you that your manuscript has been judged scientifically suitable for publication and will be formally accepted for publication once it meets all outstanding technical requirements.

Kind regards,

Ayako Mochizuki

Academic Editor

PLOS ONE

Additional Editor Comments (optional):

I am glad to say that reviewers and I are satisfied with your manuscript and have decided that it is appropriate to publish it in PLOS ONE. Congratulations on your excellent work!

Reviewers' comments:

Reviewer's Responses to Questions

**Comments to the Author**

1. If the authors have adequately addressed your comments raised in a previous round of review and you feel that this manuscript is now acceptable for publication, you may indicate that here to bypass the “Comments to the Author” section, enter your conflict of interest statement in the “Confidential to Editor” section, and submit your "Accept" recommendation.

Reviewer #1: All comments have been addressed

Reviewer #2: All comments have been addressed

2. Is the manuscript technically sound, and do the data support the conclusions?

Reviewer #1: (No Response)

Reviewer #2: Yes

3. Has the statistical analysis been performed appropriately and rigorously? 

Reviewer #1: (No Response)

Reviewer #2: Yes

4. Have the authors made all data underlying the findings in their manuscript fully available?

Reviewer #1: (No Response)

Reviewer #2: Yes

5. Is the manuscript presented in an intelligible fashion and written in standard English?

Reviewer #1: (No Response)

Reviewer #2: Yes

6. Review Comments to the Author

Reviewer #1: (No Response)

Reviewer #2: (No Response)

7. PLOS authors have the option to publish the peer review history of their article (what does this mean? ). If published, this will include your full peer review and any attached files.

**Do you want your identity to be public for this peer review?** For information about this choice, including consent withdrawal, please see our Privacy Policy .

Reviewer #1: No

Reviewer #2: **Yes: ** Kouta Nagoya

---

## [Editor Report · Acceptance letter]

PONE-D-24-55641R2

PLOS ONE

Dear Dr. Liu,

I'm pleased to inform you that your manuscript has been deemed suitable for publication in PLOS ONE. Congratulations! Your manuscript is now being handed over to our production team.

Kind regards,

on behalf of

Dr. Ayako Mochizuki

Academic Editor

PLOS ONE